# Optimizing Robustness and Accuracy in Mixture of Experts: A Dual-Model Approach

**Xu Zhang** [1]  **Kaidi Xu** [2]  **Ziqing Hu** [3]  **Ren Wang** [1]

## Abstract

Mixture of Experts (MoE) have shown remarkable success in leveraging specialized expert networks for complex machine learning tasks. However, their susceptibility to adversarial attacks presents a critical challenge for deployment in robust applications. This paper addresses the critical question of how to incorporate robustness into MoEs while maintaining high natural accuracy. We begin by analyzing the vulnerability of MoE components, finding that expert networks are notably more susceptible to adversarial attacks than the router. Based on this insight, we propose a targeted robust training technique that integrates a novel loss function to enhance the adversarial robustness of MoE, requiring only the robustification of one additional expert without compromising training or inference efficiency. Building on this, we introduce a dual-model strategy that linearly combines a standard MoE model with our robustified MoE model using a smoothing parameter. This approach allows for flexible control over the robustness-accuracy trade-off. We further provide theoretical foundations by deriving certified robustness bounds for both the single MoE and the dual-model. To push the boundaries of robustness and accuracy, we propose a novel joint training strategy JTDMoE for the dual-model. This joint training enhances both robustness and accuracy beyond what is achievable with separate models. Experimental results on CIFAR-10 and TinyImageNet datasets using ResNet18 and Vision Transformer (ViT) architectures demonstrate the effectiveness of our proposed methods. The code is publicly available at `https://github.com/TIML-Group/Robust-MoE-Dual-Model`.

[1]Illinois Institute of Technology [2]Drexel University [3]Perplexity AI. Correspondence to: Ren Wang <rwang74@iit.edu>.

*Proceedings of the 42nd International Conference on Machine Learning*, Vancouver, Canada. PMLR 267, 2025. Copyright 2025 by the author(s).

## 1. Introduction

The Mixture of Experts (MoE) architecture has emerged as a powerful framework in machine learning, enabling models to capture complex patterns by combining the strengths of multiple specialized expert networks. Originally introduced to enhance model capacity without a proportional increase in computational cost, the MoE framework operates in a straightforward but effective way: different components of the model, known as experts, specialize in distinct tasks or features of the data (Jacobs et al., 1991; Jordan & Jacobs, 1994). MoE leverage a router to dynamically assign input data to the most appropriate expert (Shazeer et al., 2017). Such mechanism makes them particularly valuable in large-scale applications such as natural language processing (Du et al., 2022) and computer vision (Riquelme et al., 2021).

Despite their success in achieving high accuracy, MoE, similar to other deep learning models, are vulnerable to adversarial attacks. Adversarial examples, which are input samples perturbed by imperceptible noise, can induce deep learning models to produce incorrect predictions with high confidence (Goodfellow et al., 2014; Carlini & Wagner, 2017; Papernot et al., 2016), posing significant risks in safety-critical applications (Finlayson et al., 2019; Li et al., 2023). The same modular structure that empowers MoE renders them particularly susceptible to adversarial attacks, as each expert presents a potential vulnerability. Our experiments in Section 4.1 confirm that targeting MoE experts can be an effective strategy for compromising the model, underscoring the need for enhanced robustness measures in MoE.

Adversarial Training (AT) and its variants have been extensively researched to defend against adversarial attacks, attracting considerable research interest (Madry et al., 2017; Zhang et al., 2019; Wang et al., 2020a; Wong et al., 2020; Wang et al., 2021; 2023). However, these methods predominantly focus on standard architectures and don't directly address the unique challenges of MoE. The heterogeneous structure of MoE, comprising a router and multiple expert networks, complicates the application of traditional robustness techniques. A few existing works have explored adversarial robustness in MoE architectures (Puigcerver et al., 2022; Zhang et al., 2023). Nonetheless, they often fail to analyze the robustness of each component and typically only

consider traditional AT or impose architectural restrictions. Moreover, the robustness enhancements in these works significantly degrade standard accuracy, limiting their practical utility.

In this paper, we aim to bridge the gap between adversarial robustness and accuracy in MoEs. We begin by examining MoE components to identify their susceptibility to adversarial attacks, finding that expert networks are more vulnerable than the router. Based on this, we propose Robust Training with Experts' Robustification (RT-ER) to enhance MoE robustness by applying robust training to the expert networks without sacrificing training or inference efficiency. Additionally, we explore combining standard and robust MoE models as dual-models to balance accuracy and robustness. We also derive theoretical robustness bounds for both the single MoE and dual-model, offering insights into achievable limits of adversarial robustness and guiding the development of a joint training strategy, JTDMoE. Our methods are illustrated in Figure 1.

Our contributions are summarized as follows:

- **Assessment of MoE Vulnerabilities and Incorporation of Robustness:** We identify key MoE components vulnerable to adversarial attacks and propose the RT-ER method, which enhances the robustness of expert networks within the MoE architecture while maintaining efficiency in both training and inference.
- **Dual-Model Strategy:** We introduce a framework combining standard and robust MoE models to balance the trade-off between accuracy and robustness.
- **Theoretical Foundations:** We derive robustness bounds for MoE and the dual-model, offering deeper insights into the capabilities and constraints of our approach.
- **Joint Training Strategy:** Leveraging our theoretical findings, we develop a joint training strategy, JTDMoE, for the dual MoE model that enhances both robustness and accuracy.

## 2. Related Work

MoE has long been explored in the machine learning community as a method for tackling complex tasks by combining specialized expert networks (Jacobs et al., 1991). Each expert focuses on certain aspects of the data, and their outputs are combined through a weighted sum determined by a gating mechanism or router (Yuksel et al., 2012). A notable advancement is the sparsely gated MoE (Shazeer et al., 2017; Wang et al., 2020b; Riquelme et al., 2021; Fedus et al., 2022; Xue et al., 2022), which activates only a subset of experts based on a routing mechanism, allowing conditional computation and enabling models to scale parameters independently of computational cost (Patterson et al., 2021).

This approach has been successfully applied in Natural Language Processing (Du et al., 2022; Lewis et al., 2021) and Computer Vision (Riquelme et al., 2021; Xue et al., 2022).

Despite their widespread application, there has been limited research on the robustness of MoE, especially in adversarial settings.Small, carefully crafted perturbations, known as adversarial examples, can cause deep neural networks to make incorrect predictions(Goodfellow et al., 2014; Carlini & Wagner, 2017; Papernot et al., 2016). Defenses against such attacks often rely on adversarial training methods based on min-max optimization (Madry et al., 2017; Zhang et al., 2019; Wang et al., 2020a; Wong et al., 2020). In the context of MoEs, only a few initial studies have begun exploring adversarial robustness with obvious limitations. The first work focused on Vision Transformers (ViTs) with MoE structures, examining the relationship between model capacity and robustness, only considering traditional adversarial training to robustify the model (Puigcerver et al., 2022). Another study investigated adversarial robustness of MoEs with a method only working on convolutional neural networks (Zhang et al., 2023). Moreover, these methods sacrifice standard accuracy for robustness, limiting their practical applications. Another line of related works is ensemble methods, which combine predictions from multiple models. Ensembles can improve robustness by aggregating predictions from multiple models, reducing the impact of individual vulnerabilities (Liu et al., 2018; Alam et al., 2022; Co et al., 2022; Bai et al., 2024a;b). While MoE models share conceptual similarities with ensembles by leveraging multiple sub-models, they differ fundamentally due to their dynamic routing mechanism, where a router assigns inputs to specific experts rather than combining outputs from all experts. This distinction necessitates tailored approaches for enhancing MoE robustness. Our work introduces a comprehensive framework to robustify MoEs while optimizing the balance between robustness and accuracy.

## 3. Preliminaries

In this section, we introduce MoE architecture considered in this paper and outline how adversarial attacks can target different components of the MoE model.

**MoE Architecture.** The MoE (Jacobs et al., 1991; Jordan & Jacobs, 1994) is a neural network architecture that leverages multiple specialized sub-networks, known as experts, to improve modeling capacity and performance on complex tasks. A router (also referred to as a gating network) determines the contribution of each expert to the final prediction based on the input data. Formally, let $E$ denote the number of experts in the MoE model. Each expert is a function $f_i(\mathbf{x})$, where $i = 1, 2, \cdots, E$, and $\mathbf{x} \in \mathbb{R}^d$ represents the input data. The router computes routing weights $a_i(\mathbf{x})$, which are typically non-negative and sum to one, i.e., $\sum_{i=1}^{E} a_i(\mathbf{x}) = 1$. The

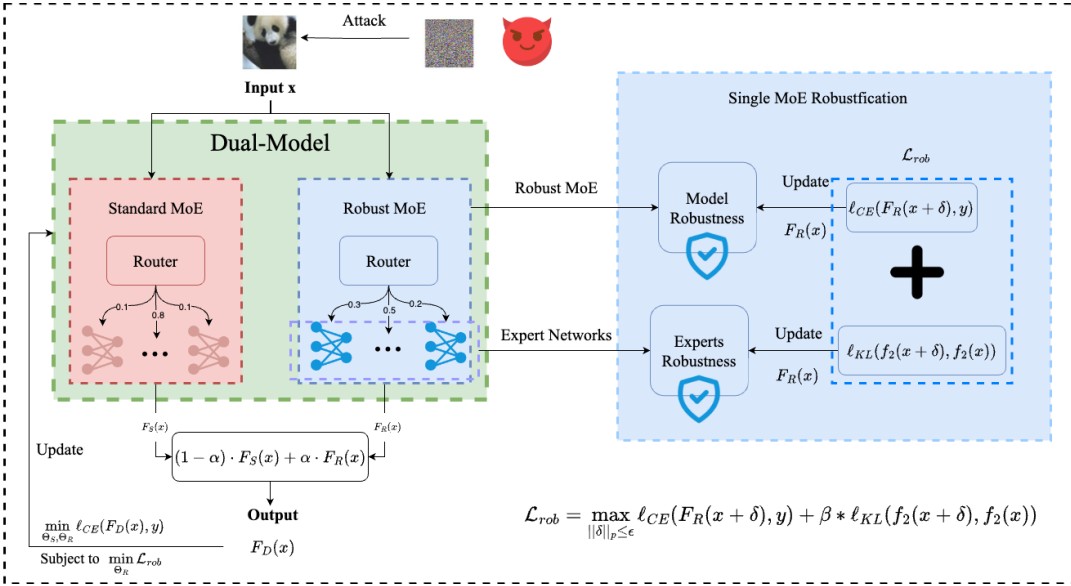

*Figure 1.* Illustration of our methods to enhancing the robustness of a single MoE and our joint training strategy for the dual-model. **Right:** Our single MoE robustification method enhances the robustness of a single MoE $F_R$ by introducing an additional term to reinforce the robustness of second-top expert $f_2$ beyond standard adversarial training. **Left:** The dual-model is a linear combination of a standard MoE $F_S$ and a robust MoE $F_R$. The jointly-trained dual-model (JTDMoE) improves robustness while maintaining high standard accuracy using a bi-level alternating training approach.

final prediction of the MoE model is given by a weighted sum of the experts' outputs $F(\mathbf{x}) = \sum_{i=1}^{E} a_i(\mathbf{x}) f_i(\mathbf{x})$. In our setup, the router $a_i(\mathbf{x})$ is implemented using a fully connected layer, and each expert is a neural network tailored to capture specific aspects of the data.

**Adversarial Attacks on MoE Models.** An adversarial attack on the entire MoE model seeks to find a perturbation $\boldsymbol{\delta}$ such that the model's output on the perturbed input $\mathbf{x} + \boldsymbol{\delta}$ differs significantly from the output on the original input $\mathbf{x}$ or its ground truth label $y$: $F(\mathbf{x} + \boldsymbol{\delta}) = \sum_{i=1}^{E} a_i(\mathbf{x} + \boldsymbol{\delta}) f_i(\mathbf{x} + \boldsymbol{\delta})$, where the $\boldsymbol{\delta}$ is usually generated by maximizing the loss function $\arg\max_{\|\boldsymbol{\delta}\|_p \le \epsilon} \mathcal{L}(F(\mathbf{x} + \boldsymbol{\delta}), y)$ under the $\ell_p$ norm and the attack budget $\epsilon$. We also consider loss terms that do not rely on $y$. Notice that the attack is on the whole MoE. We implement adversarial attacks focusing on specific components of the MoE model. An attack targeting the router aims to alter the routing weights without affecting the experts' outputs. The perturbation $\boldsymbol{\delta}$ is crafted by changing the router's decisions while the experts receiving the original input: $F(\mathbf{x} + \boldsymbol{\delta}) = \sum_{i=1}^{E} a_i(\mathbf{x} + \boldsymbol{\delta}) f_i(\mathbf{x})$. An attack targeting the experts aims to alter the experts' outputs without changing the routing weights. $\boldsymbol{\delta}$ is designed such that: $F(\mathbf{x} + \boldsymbol{\delta}) = \sum_{i=1}^{E} a_i(\mathbf{x}) f_i(\mathbf{x} + \boldsymbol{\delta})$. We remark that once the perturbation $\boldsymbol{\delta}$ is generated, it will be implemented on both the router and experts despite of its target. When attacking the expert networks, the perturbations are generated under the assumption that the routing scores remain unchanged. The generated perturbations can still alter the final predic-

tions of MoE models, as they are largely independent of the router's behavior. Empirically, our experiments show that in 98% of cases, the perturbed inputs are still routed to the same expert(s), indicating that the expert-targeted attacks remain effective despite this assumption.

# 4. Robust Training for Mixture of Experts

In this section, we aim to enhance the adversarial robustness of MoE. The key question we pose here is as follows.

> **(Q1)** *Which part of MoE is most vulnerable to attacks and how should we robustify the critical part?*

## 4.1. Assessing the Robustness of MoE Components

To enhance the robustness of MoE, it is essential to understand which component is most susceptible to adversarial attacks. We begin by assessing the robustness of both the router and expert networks in MoE when subjected to adversarial attacks. We analyze the vulnerability of these components by evaluating a standard MoE model trained with the cross-entropy loss $\arg\min_{\boldsymbol{\Theta}_S} \ell_{CE}(F_S(\mathbf{x}), y)$, where $F_S(\cdot)$ is the output of the standard MoE model $\boldsymbol{\Theta}_S$ that has only seen clean data during the training.

To specifically evaluate the accuracy and robustness of MoE, and the robustness of the router and the experts, we introduce four metrics: ❶ Standard Accuracy (**SA**) is the accuracy on clean test data; ❷ Robust Accuracy (**RA**) is the

accuracy on adversarially perturbed test data generated by attacking the whole MoE; ❸ **RA-E** is the accuracy on adversarially perturbed test data generated by attacking the experts; ❹ **RA-R** is the accuracy on adversarially perturbed test data generated by attacking the router. The RA-E/RA-R reflects the experts'/router's ability to maintain correct predictions when subjected to adversarial perturbations.

*Table 1.* Robustness assessment of MoE components on CIFAR-10 and TinyImageNet. RA-E values are small under both Projected Gradient Descent Attack (PGD) (Madry et al., 2017) and AutoAttack (AA) (Croce & Hein, 2020), indicating the vulnerability of the experts. Throughout the paper, we highlight the most vulnerable metric in bold to emphasize its susceptibility.

| CIFAR-10 | | | | |
|---|---|---|---|---|
| Attack Method | SA(%) | RA(%) | RA-E(%) | RA-R(%) |
| PGD | 92.14 | 52.54 | **3.11** | 54.67 |
| AutoAttack | 92.14 | 14.25 | **2.95** | 55.2 |
| **TinyImageNet** | | | | |
| Attack Method | SA(%) | RA(%) | RA-E(%) | RA-R(%) |
| PGD | 82.05 | 34.53 | **29.21** | 59 |
| AutoAttack | 82.05 | **14.24** | 26.11 | 58.95 |

We conduct a pilot study on CIFAR-10 and TinyImageNet datasets using MoE models with four experts. For CIFAR-10, we use ResNet18 as experts and set $\epsilon = 8/255$. For TinyImageNet, we use ViT-small as experts with $\epsilon = 2/255$. We employ a 50-step Projected Gradient Descent Attack (PGD) (Madry et al., 2017) and AutoAttack (AA) (Croce & Hein, 2020). The results are summarized in Table 1. On both datasets, the RA-R remains above 50% under both PGD and AA, suggesting that the router maintains reasonable performance under attack. In contrast, the RA-E drops below 4%/30% on CIFAR-10/TinyImageNet, indicating that the experts are highly susceptible to adversarial perturbations. This vulnerability causes the overall RA of the MoE model to be low. The experimental results indicate that the expert networks are significantly more vulnerable to adversarial attacks than the router. This is due to the fact that the experts are more complex than the router architecture. These findings highlight the importance of focusing on enhancing the robustness of the expert networks to improve the overall resilience of MoE models against adversarial attacks.

## 4.2. Incorporating Robustness into MoE Architecture

To incorporate robustness into the MoE architecture, a straightforward approach is to apply traditional adversarial training (Madry et al., 2017), which involves training the model on adversarial examples generated during training. The traditional adversarial training is defined as:

$$\min_{\boldsymbol{\Theta}_R} \max_{\|\boldsymbol{\delta}\|_p \leq \epsilon} \ell_{CE}(F_R(\mathbf{x} + \boldsymbol{\delta}), y), \qquad (1)$$

where $F_R(\cdot)$ is the output of the robust MoE model $\boldsymbol{\Theta}_R$ under robust training. However, when applied to MoE models, traditional adversarial training exhibits several issues: (**I1**) **Low Robust Accuracy:** The final RA remains unacceptably low. In our experiments on CIFAR-10 with ResNet18 experts, the model achieves 79.08% SA and only 53.74% RA on the test set using PGD after 130 training epochs. (**I2**) **Training Instability:** The training process is unstable, with sudden drops in both SA and RA. As shown in Figure 2, both metrics decline sharply between epochs 80 and 90. This is due to the router selecting a different expert as the primary contributor when faced with adversarially perturbed inputs. Since standard adversarial training primarily focuses on optimizing the robustness of the overall model output, it may overlook individual experts' robustness, resulting in some experts remaining vulnerable to attacks. Consequently, when the router routes to one of these weaker experts, the model's robustness is compromised. These issues suggest that traditional adversarial training is not well-suited for MoE architectures due to the complex interplay between the router and expert networks.

Based on our earlier assessment (in Section 4.1) that the experts are the most vulnerable components, we propose a new robust training approach that specifically targets enhancing the robustness of the expert networks. Our approach modifies the loss function to include a Kullback-Leibler (KL) divergence term that encourages the experts' outputs on adversarial examples to be similar to their outputs on clean inputs. We propose a Robust Training with Expert Robustification (RT-ER) approach:

$$\min_{\boldsymbol{\Theta}_R}[\mathcal{L}_{rob} = \max_{\|\boldsymbol{\delta}\|_p \leq \epsilon} \ell_{CE}(F_R(\mathbf{x} + \boldsymbol{\delta}), y) \\ + \beta \cdot \ell_{KL}(f_2(\mathbf{x} + \boldsymbol{\delta}), f_2(\mathbf{x}))], \qquad (2)$$

where $f_2(\mathbf{x})$ denotes the output of the expert with the second-largest router weight for the adversarial example. The term $\ell_{KL}(f_2(\mathbf{x} + \boldsymbol{\delta}), f_2(\mathbf{x}))$ represents the KL divergence between the output of this expert on the adversarial example and its output on the clean input. RT-ER is compatible with various routing strategies. For instance, when the router employs a top-$n$ strategy, RT-ER selects an additional expert—one not initially chosen by the router—based on the router weights, leveraging the router's output. We adopt KL divergence instead of cross-entropy for two primary reasons. First, cross-entropy relies on a well-defined ground-truth target distribution, which is unavailable in our setting. Second, employing cross-entropy in the second term would compel the predictions to closely match the ground truth, potentially resulting in overfitting. In contrast, KL divergence enables a softer alignment between experts, thereby reducing the risk of overfitting and offering greater flexibility in modeling uncertainty. The hyperparameter $\beta$ controls the trade-off between MoE-wide robustness and

expert-specific robustness. For clarity, we refer to the expert with the second-largest router weight as the second-top expert in the following discussion.

The concept of RT-ER is illustrated in the right panel of Figure 1. By incorporating the KL divergence term, we explicitly encourage second-top expert to produce similar outputs for both clean and adversarial inputs. RT-ER offers several advantages: (**A1**) By penalizing deviations in experts' outputs due to adversarial perturbations, we strengthen the experts against attacks and thus contributes to better overall RA for the MoE model. (**A2**) The inclusion of the KL divergence term helps stabilize training by regularizing the experts' behavior. When second-top expert is also robust to adversarial inputs, the model maintains greater stability, even when the router activates different experts. This enhanced consistency reduces performance fluctuations under adversarial attacks, allowing the model to sustain stronger performance on adversarial samples. By ensuring all experts are robust, the model avoids scenarios where only a subset of experts is resilient, leaving others vulnerable. This uniform robustness across experts leads to a more reliable defense against adversarial perturbations, improving the model's overall resilience. The effectiveness of our proposed robust training method is demonstrated through the experimental results presented in Section 6.2. We observe significant improvements in RA compared to traditional adversarial training and a more stable training process, validating the efficacy of focusing on expert robustness in MoE models.

*Table 2.* Robustness evaluation of adversarial training (AT) and our method (RT-ER) under PGD attacks across different routing strategies. RT-ER consistently achieves higher robust accuracy (RA) than AT under both top-1 and top-2 routing strategies.

| Top-1 Strategy | | | | |
| --- | --- | --- | --- | --- |
| Method | SA(%) | RA(%) | RA-E(%) | RA-R(%) |
| AT | 79.08 | **53.74** | 73.85 | 78.91 |
| RT-ER | 77.81 | **69.09** | 75.71 | 72.28 |
| Top-2 Strategy | | | | |
| Method | SA(%) | RA(%) | RA-E(%) | RA-R(%) |
| AT | 79.20 | **54.65** | 74.66 | 76.09 |
| RT-ER | 78.78 | **70.05** | 76.2 | 72.35 |

Compared to AT, our proposed method, RT-ER, is more efficient and adaptable to various routing strategies, as demonstrated in Table 2. We evaluate robustness under both the top-1 and top-2 routing strategies, where the router activates one or two experts, respectively. First, AT, as an end-to-end training approach, proves insufficient for robustifying MoE architectures. Even with AT applied, the MoE model achieves only around 54% RA, whereas RT-ER improves RA by more than 16%. Second, RT-ER effectively enhances robustness across different routing strategies, consistently outperforming AT under both top-1 and top-2 routing. More-

over, RT-ER requires training only one additional expert, introducing minimal overhead to training efficiency. Further robustness evaluations under smooth attacks on MoE are provided in Appendix A.3. In summary, RT-ER not only strengthens the robustness of MoE architectures but also preserves both training and inference efficiency.

## 5. Dual-Model Strategy for Robustness and Accuracy

While our proposed RT-ER method enhances robustness against adversarial attacks (measured by RA), it may inadvertently degrade performance on clean data (measured by SA). This prompts the following research question:

> (**Q2**) *How can we incorporate robustness into Mixture of Experts while minimizing the robustness-accuracy trade-off?*

### 5.1. Dual-Model with Post-Training MoE

Traditional MoE models are efficient but vulnerable to attacks, while robust MoEs withstand attacks but often reduce standard accuracy. We explore whether combining both models can balance robustness and accuracy. Let $F_S(\mathbf{x})$ be a standard MoE and $F_R(\mathbf{x})$ a robust MoE. The dual-model, $F_D(\mathbf{x})$, is defined as:

$$F_D(\mathbf{x}) = (1 - \alpha) \cdot F_S(\mathbf{x}) + \alpha \cdot F_R(\mathbf{x}), \qquad (3)$$

where $\alpha$ controls the robustness-accuracy trade-off. In general, the standard MoE exhibits higher SA, while the robust MoE exhibits higher RA. The parameter $\alpha$ controls the contribution of the robust MoE to the final prediction. As $\alpha$ increases, the robust MoE contributes more to the final prediction, enhancing adversarial robustness but potentially degrading performance on clean data. We set $\alpha \geq 0.5$ to ensure that the dual-model remains robust.

Although the dual-model structure helps balance performance on clean and adversarial data, it introduces additional complexity in understanding its robustness. To clarify the influence of each parameter on robustness, we derive a certified robustness bound for the dual model $F_D$.

We first provide the definition of certified robustness in MoE under the perturbation bounded by $\ell_p$ norm[1]

**Definition 5.1.** Consider a robust MoE $F_R : \mathbb{R}^d \to \mathbb{R}^c$ and an arbitrary input $\mathbf{x} \in \mathbb{R}^d$. Let $y = \arg\max_i F_R(\mathbf{x})$, with bound $\epsilon \geq 0$. We say $F_R$ is certifiably robust at $\mathbf{x}$ with bound $\epsilon$ if for all $k \neq y$ and $\boldsymbol{\delta} \in \mathbb{R}^d$ such that $\|\boldsymbol{\delta}\|_p \leq \epsilon$, the following holds:

$$F_R^{(y)}(\mathbf{x} + \boldsymbol{\delta}) \geq F_R^{(k)}(\mathbf{x} + \boldsymbol{\delta}) \qquad (4)$$

---

[1]While our theorems hold for $p \in [1, \infty)$, we implement $p = \infty$ in all our experiments.

This definition formalizes the certifiable robustness of $F_R$ at $x$, ensuring that the model's top prediction remains consistent under perturbations within an $\ell_p$-norm ball of radius $\epsilon$. For practical classifiers, the robust margin $F_R^{(y)}(\mathbf{x}+\boldsymbol{\delta}) - F_R^{(k)}(\mathbf{x}+\boldsymbol{\delta})$ can be estimated by evaluating the confidence gap between predicted and runner-up classes on a strong adversarial input.

**Definition 5.2.** A function $f : \mathbb{R}^d \to \mathbb{R}$ is $\ell_p$-Lipschitz continuous if there exists $L \in (0, \infty)$ such that for all $\mathbf{x}', \mathbf{x} \in \mathbb{R}^d$, $|f(\mathbf{x}') - f(\mathbf{x})| \leq L\|\mathbf{x}' - \mathbf{x}\|_p$. The Lipschitz constant of $f$ is defined as

$$Lip_p(f) := \inf\{L : |f(\mathbf{x}') - f(\mathbf{x})| \leq L\|\mathbf{x}' - \mathbf{x}\|_p\} \quad (5)$$

Following the definition of $\ell_p$-Lipschitz continuity, we can introduce the following assumption:

**Assumption 5.3.** Each expert and the router in the robust MoE $F_R(\mathbf{x})$ is $\ell_p$-Lipschitz continuous, where the experts' Lipschitz continuity is given by

$$|f_{R_i}^{(y)}(\mathbf{x}+\boldsymbol{\delta}) - f_{R_i}^{(y)}(\mathbf{x})| \leq L_{R_i}\|\boldsymbol{\delta}\|_p \quad (6)$$

and the router's Lipschitz continuity at $i$-th output by

$$|a_{R_i}(\mathbf{x}+\boldsymbol{\delta}) - a_{R_i}(\mathbf{x})| \leq r_{R_i}\|\boldsymbol{\delta}\|_p \quad (7)$$

Our theoretical results are all based on Assumption 5.3, which constrains the applicability of our method. Therefore, it is important to clarify the scope under which this assumption holds. We identify three representative scenarios where Assumption 5.3 naturally applies. First, in sparse MoE after robust training, robust optimization techniques typically encourage large margins in expert scores. This leads to a clear separation between the top-n experts and the rest, making it difficult for small adversarial perturbations to change the top-n set. As a result, the router becomes locally stable and selects the same expert(s) for both clean and perturbed inputs. Second, in dense MoE (Zhang et al., 2024), all experts are activated and the routing function is continuous, which trivially satisfies the assumption. Third, soft MoE (Puigcerver et al., 2023) also employs a continuous routing function with soft assignments, ensuring that small input changes lead to smooth changes in routing outputs. In all these cases, the routing behavior is stable under small perturbations, thus meeting the requirement of Assumption 5.3.

**Theorem 5.4.** *Under Assumption 5.3, let $M_{R_i} \leq 1$ be an upper bound on $f_{R_i}^{(y)}(\mathbf{x})$ for any input $\mathbf{x} \in \mathbb{R}^d$. Then the robustness bound $\epsilon$ for $F_R(\mathbf{x})$ is:*

$$\epsilon = \min_{k \neq y} \frac{F_R^{(y)}(\mathbf{x}) - F_R^{(k)}(\mathbf{x})}{\sum_i (r_{R_i} M_{R_i} + a_{R_i}(\mathbf{x}) L_{R_i})}, \quad (8)$$

where $y$ is the true label and $k$ represents other classes. From Equation (8), we observe that when the experts are complex

neural networks, the robustness of the robust MoE is primarily determined by the Lipschitz constants of its individual experts. If any expert lacks robustness, the output of the vulnerable expert will vary significantly when handling adversarial inputs, indicating a large Lipschitz constant $L_{R_i}$. Consequently, the achievable certified robustness bound $\epsilon$ for the robust MoE becomes smaller. This implies that, to enhance the robustness of the robust MoE, it is essential to robustify each expert. This conclusion provides theoretical support for our proposed RT-ER method.

Motivated by Theorem 3.5 in (Bai et al., 2024a), we derive a certified bound to provide a formal guarantee on the dual-model's resistance to adversarial perturbations, stated as follows:

**Theorem 5.5.** *For a dual-model $F_D(\mathbf{x})$ comprising a standard MoE $F_S(\mathbf{x})$ and a robust MoE $F_R(\mathbf{x})$, with smoothing parameter $\alpha \in [\frac{1}{2}, 1]$, we have that $F_D(\mathbf{x}+\boldsymbol{\delta}) = y$ for all $\boldsymbol{\delta} \in \mathbb{R}^d$ such that*

$$\|\boldsymbol{\delta}\|_p \leq \epsilon = \min_{k \neq y} \frac{\alpha(F_R^{(y)}(\mathbf{x}) - F_R^{(k)}(\mathbf{x})) + \alpha - 1}{\alpha \sum_i \left(2r_{R_i} + a_{R_i}(\mathbf{x})(L_{R_i}^{(y)} + L_{R_i}^{(k)})\right)} \quad (9)$$

The proofs for Theorems 5.4 and 5.5 are provided in Appendix A.1. Theorem 5.5 demonstrates that the dual-model's robustness is fundamentally rooted in the robust MoE, and increasing the margin $F_R^{(y)}(\mathbf{x}) - F_R^{(k)}(\mathbf{x})$ could enhance the overall robustness bound. Additionally, each expert in the robust MoE influences both the robust MoE and the dual-model, underscoring the critical role of fortifying each expert's robustness. This also supports the use of our proposed loss function in Equation (2), designed to reinforce the robustness of individual experts in the MoE.

Based on Theorem 5.5, there are three ways to further enhance the robustness of the dual-model: increasing the value of $\alpha$, enlarging the margin, and decreasing the Lipschitz constant of both the experts and the router. The maximum value of $F_R^{(y)}(\mathbf{x}) - F_R^{(k)}(\mathbf{x})$ is 1. If $\alpha$ falls below 0.5, the numerator in the certified robustness bound becomes negative, rendering the robustness radius undefined. Therefore, to ensure that the dual-model maintains a valid certified robustness guarantee, $\alpha$ must lie within the range $[0.5, 1]$. Furthermore, a higher value of $\alpha$, which implies a larger contribution of the robust MoE to the final prediction, would result in a decrease in SA, which contradicts the motivation for using the dual-model architecture. As for the Lipschitz constants, they have already been optimized during RT-ER. In summary, the most efficient way to enhance the dual-model's performance is by enlarging its margin. Additionally, a larger margin also leads to better accuracy on clean data.

After analyzing the implications of the theorem, it becomes evident that the dual-model must excel as a unified system.

This observation raises the following research question:

> **(Q3)** *Can we further enhance performance by employing a joint training strategy?*

### 5.2. Boost Dual-Model with Joint Training

Inspired by Theorem 5.5 and the insights from Section 5.1, we explore the joint training strategy that considers both the dual-model and single components. By jointly training, we aim to increase the robust MoE's margin, thereby boosting both robustness and accuracy. Traditional adversarial training methods focus primarily on the overall model robustness, often overlooking the individual robustness of each MoE component, making these methods less effective in improving MoE-specific robustness.

To address this gap, we introduce a novel adversarial training framework for the dual-model based on bi-level optimization. It provides a hierarchical learning approach, where the upper-level objective depends on the solution of the lower-level problem. Specifically, it is formulated as follows:

$$\min_{\boldsymbol{\Theta}_S, \boldsymbol{\Theta}_R} \ell_{CE}(F_D(\mathbf{x}), y) \quad \text{subject to} \quad \min_{\boldsymbol{\Theta}_R} \mathcal{L}_{rob}, \quad (10)$$

where $\mathcal{L}_{rob}$ denotes the proposed loss function in Equation (2). Here, the dual-model parameters are divided into variables $\boldsymbol{\Theta}_R$ for the robust MoE and variables $\boldsymbol{\Theta}_S$ for the standard MoE. We term this new approach Joint Training for Dual-Model based on MoE (JTDMoE). JTDMoE promotes alignment between the standard MoE and robust MoE, ultimately enhancing the dual-model's performance. The complete training process is outlined in Algorithm 1.

---

**Algorithm 1** The JTDMoE algorithm

---

1: **Initialize:** robust MoE parameters $\boldsymbol{\Theta}_R$, standard MoE parameters $\boldsymbol{\Theta}_S$, batch size $\lfloor$, and attack step count $K$.
2: **for** iteration $t = 0, 1, \dots$ **do**
3:   Lower-level $\boldsymbol{\Theta}_R$-update: update $\boldsymbol{\Theta}_R$ by minimizing $\mathcal{L}_{rob}$ using a $K$-step PGD attack on batch $\mathcal{B}$.
4:   Upper-level $\boldsymbol{\Theta}_S, \boldsymbol{\Theta}_R$-update: update $\boldsymbol{\Theta}_S$ and $\boldsymbol{\Theta}_R$ by minimizing $\ell_{CE}(F_D(\mathbf{x}), y)$ on batch $\mathcal{B}$.
5: **end for**

---

The reason we use this bi-level alternating training approach is to enhance the dual-model's accuracy on clean data while maintaining its robustness as much as possible. For the lower-level optimization problem, we use RT-ER to robustify the robust MoE, as the dual-model's robustness derives from the robust MoE (Theorem 5.5). This approach leads to smaller Lipschitz constants for the experts and router, thus guaranteeing the dual-model's robustness. For the upper-level optimization problem, our goal is to enhance the dual-model's SA by minimizing $\ell_{CE}(F_D(\mathbf{x}), y)$. Additionally, the margin of the robust MoE can increase as a

side effect, further improving the dual-model's robustness. Based on this analysis, we argue that the dual-model can enhance its performance in both SA and RA through the JTDMoE approach.

## 6. Experiments

In this section, we present the effectiveness of our proposed RT-ER and the JTDMoE approach.

### 6.1. Experiment Setup

**Datasets and Model Architectures.** Our experiments use the MoE architecture on CIFAR-10 (Krizhevsky et al., 2009) and TinyImageNet (Deng et al., 2009), with a fully connected top-1 router and 4 experts ($E = 4$). We use ResNet18 (He et al., 2016) for CIFAR-10 and pre-trained ViT-small (Liu et al., 2021) for TinyImageNet. Instead of training the dual-model from scratch, we apply the JTDMoE algorithm to pre-trained models for efficiency, using ST MoE as the standard and RT-ER MoE as the robust MoE, with $\alpha$ ranging from 0.5 to 1. More details can be found in Appendix A.2.

**Robustness Evaluation.** We use PGD (Madry et al., 2017) and AutoAttack (Croce & Hein, 2020) to assess model performance under adversarial conditions, with $\epsilon = 8/255$ for CIFAR-10 and $\epsilon = 2/255$ for TinyImageNet. We train ResNet18-based MoE for 130 epochs on CIFAR-10 and fine-tune pre-trained ViT-small-based MoE for 10 epochs on TinyImageNet. A Cyclic Learning Rate strategy (Smith, 2017), starting at 0.0001, and data augmentation (Rebuffi et al., 2021) are used to enhance performance. Evaluation is done using either a 50-step PGD or AutoAttack with the same step size.

**Baseline Methods.** For comparative analysis, we define three approaches for the single MoE: (1) ST: Standard training on MoE. (2) AT: Adversarial training on MoE (Madry, 2017). (3) RT-ER: Robust training with experts robustification on MoE. In addition we cover (4) TRADES (Zhang et al., 2019) and (5) AdvMoE (Zhang et al., 2023) (a new router-expert alternating Adversarial training framework for MoE) in the Appendix A.3.

### 6.2. Evaluation of RT-ER

To showcase the improved robustness and training stability of RT-ER, we compare its SA and RA to those of AT during training. The experimental results of MoE-Resnet18 on CIFAR-10 dataset are illustrated in Figure 2. RT-ER demonstrates greater stability compared to traditional adversarial training (AT). In the case of AT-trained MoE, both SA and RA drop significantly—by over 20% and 10%, respectively—between epochs 80 and 90. Although the ViT-small

*Table 3.* Evaluation of our proposed methods, RT-ER, Dual-Model (Pretrained), and Dual-Model (JTDMoE), compared with Standard Training (ST) and Adversarial Training (AT) (Madry et al., 2018) on the CIFAR-10 and TinyImageNet datasets. We report standard accuracy (SA), robust accuracy against attacks on the entire model (RA), on experts (RA-E), and on the router (RA-R) using a 50-step PGD attack (Madry et al., 2018) in the single MoE scenario. The RT-ER method achieves the highest RA, RA-E, and RA-R among single MoE models. For dual-model scenarios, RA-R and RA-E are replaced with robust accuracy against attacks on the robust MoE (RA-RMoE) and on the standard MoE (RA-SMoE). The results show that the Dual-Model improves upon RT-ER in the accuracy-robustness trade-off, and our JTDMoE method outperforms the Dual-Model (Pre-trained) under the same smoothing parameter $\alpha$.

| CIFAR-10 | | | | TinyImageNet | | | | |
|---|---|---|---|---|---|---|---|---|
| Single MoE Performance | | | | | | | | |
| Method | SA(%) | RA(%) | RA-E(%) | RA-R(%) | Method | SA(%) | RA(%) | RA-E(%) | RA-R(%) |
| ST | 92.14 | 52.54 | **3.11** | 54.67 | ST | 82.05 | 34.53 | **29.21** | 59.00 |
| AT | 79.08 | **53.74** | 73.85 | 78.91 | AT | 70.77 | 50.94 | **42.09** | 83.81 |
| RT-ER | 77.81 | **69.09** | 75.71 | 72.28 | RT-ER | 68.51 | 56.79 | **46.32** | 81.26 |
| Dual-Model (Pre-trained) | | | | | | | | |
| $\alpha$ | SA(%) | RA(%) | RA-RMoE(%) | RA-SMoE(%) | $\alpha$ | SA(%) | RA(%) | RA-RMoE(%) | RA-SMoE(%) |
| 0.7 | 89.81 | 67.81 | 79.47 | **52.93** | 0.7 | 80.74 | 53.11 | 68.92 | **47.39** |
| 0.8 | 87.58 | 68.27 | 77.25 | 54.99 | 0.8 | 77.89 | 55.27 | 65.94 | 55.16 |
| 0.9 | 83.27 | 70.73 | 75.03 | 60.87 | 0.9 | 74.2 | 56.58 | 61.95 | 62.80 |
| Dual-Model (JTDMoE) | | | | | | | | |
| $\alpha$ | SA(%) | RA(%) | RA-RMoE(%) | RA-SMoE(%) | $\alpha$ | SA(%) | RA(%) | RA-RMoE(%) | RA-SMoE(%) |
| 0.7 | 92.29 | 74.62 | 85.42 | **68.73** | 0.7 | 84.91 | 57.39 | 69.58 | **53.32** |

MoE model shows some fluctuations as well, the variation is less pronounced, due to the robustness of the pre-trained ViT-small experts used. The experimental results of ViT-small are presented in Appendix A.3. In terms of RA, our method consistently outperforms AT across all epochs, achieving over 15.35% and 5.5% improvements in final performance on CIFAR-10 and TinyImageNet, respectively. This suggests that RT-ER is a more effective approach for enhancing the robustness of MoE architectures. Overall, our method enables faster, more efficient robustness enhancement for MoE models while reducing SA by $< 1.3\%$.

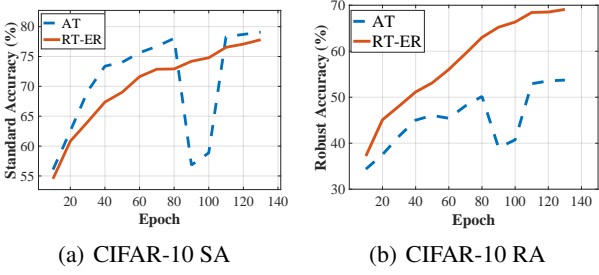

(a) CIFAR-10 SA          (b) CIFAR-10 RA

*Figure 2.* Performance evaluation of AT MoE and RT-ER MoE models with ResNet18 on the CIFAR-10 test dataset. We report standard accuracy (SA) and robust accuracy (RA) under a 50-step PGD attack, using models trained with a 10-step PGD attack. Our results indicate that RT-ER achieves consistently higher RA and demonstrates greater stability than AT MoE. For a comparable analysis using ViT-small, please refer to Appendix A.3.

In the Single MoE Performance section of Table 3, we compare ST, AT, and RT-ER using ResNet18 on CIFAR-10

and ViT-small on TinyImageNet under PGD attack. RT-ER improves RA-E by over 70% and 17% on ResNet18 and ViT-small, respectively, compared to ST, and achieves 5.8% and 4.2% improvements in RA and RA-E over AT using ViT-small. These results show RT-ER's effectiveness in robustifying single MoE models. For a single MoE, robustness is measured by the minimum of RA, RA-E, RA-R, reflecting the worst-case performance. RT-ER outperforms AT and ST in this regard. We also compare our method with TRADES (Zhang et al., 2019) and AdvMoE (Zhang et al., 2023), and examine the effect of expert number on RT-ER. Details are in Appendix A.3.

### 6.3. Evaluation of the Dual-Model Based on Pre-Trained MoE

The pre-trained dual-model combines a standard MoE and a robust MoE with a fixed $\alpha$ to improve clean data performance. On CIFAR-10, the standard MoE achieves 92.14% SA and 52.54% RA, while the robust MoE reaches 77.81% SA and 69.09% RA. On TinyImageNet, the standard MoE achieves 82.05% SA and 34.53% RA, and the robust MoE reaches 68.51% SA and 56.79% RA. As $\alpha$ increases, SA decreases but RA improves, as the robust MoE's contribution grows. When $\alpha = 1$, the dual-model becomes RT-ER, with only the robust MoE. Results for $\alpha = 0.7$, 0.8, and 0.9 under PGD attack show that increasing $\alpha$ decreases SA (by 6.54% and 13.54% on CIFAR-10 and TinyImageNet, respectively) but improves RA (by 2.92% and 3.47%, respectively) and RA-SMoE (by 7.94% and 15.41%).

Specifically, when adversarial perturbations target the robust MoE, they have minimal effect on the standard MoE,

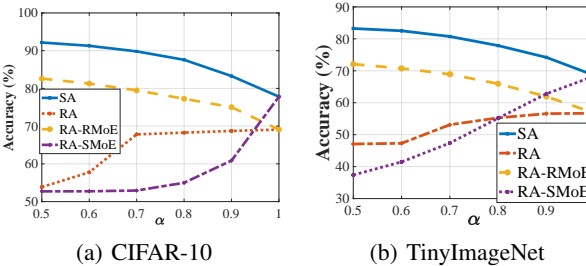

(a) CIFAR-10      (b) TinyImageNet

*Figure 3.* Performance evaluation of the Dual-Model using pre-trained MoE models. We assess the performance of the Dual-Model, which combines a standard MoE (ST) and a robust MoE (RT-ER) from Table 3. The weighting parameter $\alpha$ is incremented from 0.5 to 1.0 in steps of 0.1; at $\alpha = 1$, the Dual-Model relies exclusively on the robust MoE. All other configurations are consistent with those detailed in Figure 2.

so RA-RMoE decreases as $\alpha$ increases. Conversely, when the standard MoE is attacked, the robust MoE is minimally impacted, leading to an increase in RA-SMoE with $\alpha$. However, when $\alpha$ shifts from 0.5 to 0.7, RA-SMoE shows only slight improvement, suggesting the robust MoE lacks confidence in its predictions and can be influenced by the standard MoE. This motivates our joint training approach to improve dual-model performance. Further analysis of the dual-model's performance with pre-trained MoE under AutoAttack is in Appendix A.4.

## 6.4. Evaluation of JTDMoE

*Table 4.* Performance evaluation of JTDMoE versus the baseline (Pre-trained Dual-Model) when attacked by AutoAttack. In the case of $\alpha = 0.7$, we report standard accuracy (SA), robust accuracy (RA), robust accuracy of the robust MoE (RA-RMoE), and robust accuracy of the standard MoE (RA-SMoE) using AutoAttack (AA) (Croce & Hein, 2020). Results demonstrate that JTDMoE consistently outperforms the baseline across all metrics.

| CIFAR-10 | | | | |
|---|---|---|---|---|
| Method | SA(%) | RA(%) | RA-RMoE(%) | RA-SMoE(%) |
| Pre-trained | 89.81 | 33.64 | 34.20 | 14.34 |
| JTDMoE | **92.29** | **54.55** | **51.13** | **45.55** |
| **TinyImageNet** | | | | |
| Method | SA(%) | RA(%) | RA-RMoE(%) | RA-SMoE(%) |
| Pre-trained | 80.74 | 51.27 | 56.25 | 26.88 |
| JTDMoE | **84.91** | **52.91** | **60.98** | **28.82** |

To compare with the results in Section 6.3, we use the same standard MoE and robust MoE models for $\alpha = 0.7$. As shown in Table 3 and Table 4, our JTDMoE algorithm significantly improves the performance of the pre-trained dual-model across all evaluation metrics. Specifically, it increases SA by 2.48% and 4.17%, and RA by 6.81% and 4.28% under PGD attacks for CIFAR-10 and TinyImageNet, respectively. When evaluated with AutoAttack, JTDMoE achieves

a 19.91% improvement in RA for CIFAR-10 and 1.64% for TinyImageNet. Additionally, JTDMoE outperforms the baseline in both RA-RMoE and RA-SMoE, further validating the effectiveness of aligning the standard and robust MoE models. These improvements are consistent with our theoretical expectations and support Theorem 5.5. Further analysis of margin improvement and additional experiments with different $\alpha$ values can be found in Appendix A.5.

**Margin Comparison.** Based on Theorem 5.5, increasing the margin $F_R^{(y)}(\mathbf{x}) - F_R^{(k)}(\mathbf{x})$ enhances the robustness of the dual-model. To validate this, we compare the margins between JTDMoE and the pre-trained dual-model under $\alpha = 0.7$ on the CIFAR-10 dataset. The results are summarized in Table 5.

*Table 5.* Margin comparison of JTDMoE and pre-trained dual-model on the CIFAR-10 test dataset. Using the pre-trained dual-model as the baseline, we report margin improvements for each class. Results show that JTDMoE consistently improves margins across all classes, supporting Theorem 5.5.

| CIFAR-10 | | | | |
|---|---|---|---|---|
| Class | Airplane | Automobile | Bird | Cat | Deer |
| Improvement | 5.11% | 3.49% | 7.5% | 5.55% | 10.27% |
| Class | Dog | Frog | Horse | Ship | Truck |
| Improvement | 14.42% | 6.46% | 4.03% | 1.25% | 4.58% |

As shown in Table 5, JTDMoE improves the margin for every class, with improvements ranging from 1.25% to 14.42%. The class Dog exhibits the most significant margin improvement of 14.42%, followed by Deer with a 10.27% improvement. These results empirically verify Theorem 5.5, demonstrating that margin enhancement contributes to JTDMoE's superior robust accuracy (RA).

## 7. Conclusion

This work presents a comprehensive framework for enhancing adversarial robustness in MoE models. Our approach is motivated by the key observation that expert networks in MoEs are significantly more susceptible to adversarial perturbations than the router—a structural vulnerability specific to the MoE architecture. To address this, we propose a targeted robustification strategy with accompanying theoretical robustness bounds, which adversarially trains an additional expert not selected by the router. Building upon the theoretical robustness–accuracy trade-off, we further introduce a dual-model framework (JTDMoE) that integrates a standard MoE and a robust MoE via a bi-level joint training scheme, achieving strong adversarial robustness without compromising standard accuracy. Extensive experiments across diverse models and datasets validate the effectiveness and scalability of our proposed methods.

## Impact Statement

This paper presents work aimed at advancing the field of Machine Learning, specifically in improving the adversarial robustness of Mixture of Experts (MoE) models. Our contributions enhance the security and reliability of MoEs, which are widely used in large-scale and specialized AI applications. While this research primarily focuses on technical advancements, its broader implications include improving the deployment of MoE-based models in safety-critical domains. However, we do not identify any specific societal consequences that must be highlighted here.

## Acknowledgement

This work is supported by the NSF under Grants 2246157 and 2319243. We are thankful for the computational resources made available through NSF ACCESS and Argonne Leadership Computing Facility.

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

# A. Supplementary Material

In this supplementary material, we provide the proofs of Theorems 5.4 and 5.5 in Section A.1. Additional experimental results are organized as follows: single MoE experiments are reported in Section A.3, dual-model experiments based on pre-trained MoEs are presented in Section A.4, and JTD-MoE experiments are detailed in Section A.5. In addition, Section A.6 presents RT-ER experiments conducted using a large model (ViT) on the large-scale ImageNet dataset.

## A.1. Proof of Key Theorems

**Proof of Theorem 5.4:** When the input is perturbed from $\mathbf{x}$ to $\mathbf{x} + \boldsymbol{\delta}$, the change in the final output of the MoE can be expressed as:

$$
\begin{aligned}
\Delta =& F_R^{(y)}(\mathbf{x} + \boldsymbol{\delta}) - F_R^{(y)}(\mathbf{x}) \\
=& \sum_i \left[ a_{R_i}(\mathbf{x} + \boldsymbol{\delta}) f_{R_i}^{(y)}(\mathbf{x} + \boldsymbol{\delta}) - a_{R_i}(\mathbf{x}) f_{R_i}^{(y)}(\mathbf{x}) \right]
\end{aligned}
$$

We decompose $\Delta$ into two terms:

$$
\begin{aligned}
\Delta_1 &= \sum_i \left( a_{R_i}(\mathbf{x} + \boldsymbol{\delta}) - a_{R_i}(\mathbf{x}) \right) f_{R_i}^{(y)}(\mathbf{x} + \boldsymbol{\delta}) \\
\Delta_2 &= \sum_i a_{R_i}(\mathbf{x}) \left( f_{R_i}^{(y)}(\mathbf{x} + \boldsymbol{\delta}) - f_{R_i}^{(y)}(\mathbf{x}) \right) \\
\Delta &= \Delta_1 + \Delta_2
\end{aligned}
$$

By this decomposition, $\Delta_1$ captures the change due to the router's output, while $\Delta_2$ represents the change due to the experts' outputs. To derive a certified bound for the overall MoE, we need to bound both $\Delta_1$ and $\Delta_2$.

**Bounding $\Delta_1$:**

$$
\begin{aligned}
\Delta_1 &\le \sum_i |a_{R_i}(\mathbf{x} + \boldsymbol{\delta}) - a_{R_i}(\mathbf{x})| \cdot f_{R_i}^{(y)}(\mathbf{x} + \boldsymbol{\delta}) \\
&\le \sum_i r_{R_i} \|\boldsymbol{\delta}\| M_{R_i},
\end{aligned}
$$

where $M_{R_i} \le 1$ is an upper bound on $f_{R_i}^{(y)}(\mathbf{x} + \boldsymbol{\delta})$ or an upper bound on $f_{R_i}^{(y)}(\mathbf{x})$ for any inputs and $r_{R_i}$ is the Lipschitz constant of the router $a_{R_i}$.

**Bounding $\Delta_2$:**

$$
\begin{aligned}
\Delta_2 &\le \sum_i a_{R_i}(\mathbf{x}) |f_{R_i}^{(y)}(\mathbf{x} + \boldsymbol{\delta}) - f_{R_i}^{(y)}(\mathbf{x})| \\
&\le \sum_i a_{R_i}(\mathbf{x}) L_{R_i} \|\boldsymbol{\delta}\|
\end{aligned}
$$

where $L_{R_i}$ is the Lipschitz constant of the expert $f_{R_i}^{(y)}$.

**Bounding $\Delta$:** Combining the bounds for $\Delta_1$ and $\Delta_2$, we have:

$$
\Delta \le \sum_i (r_{R_i} M_{R_i} + a_{R_i}(\mathbf{x}) L_{R_i}) \|\boldsymbol{\delta}\|
$$

By defining the total Lipschitz constant $L_{total}$ as

$$
L_{total} = \sum_i (r_{R_i} M_{R_i} + a_{R_i}(\mathbf{x}) L_{R_i}),
$$

the upper bound on the change in the final output becomes:

$$
\left| F_{R_i}^{(y)}(\mathbf{x} + \boldsymbol{\delta}) - F_{R_i}^{(y)}(\mathbf{x}) \right| \le L_{total} \|\boldsymbol{\delta}\|
$$

To ensure the prediction remains unchanged under perturbation $\boldsymbol{\delta}$, the change in output must not exceed the classification margin $m = \min_{k \ne y} F_R^{(y)}(\mathbf{x}) - F_R^{(k)}(\mathbf{x})$. Thus:

$$
\begin{aligned}
L_{total} \|\boldsymbol{\delta}\| &\le m \\
\|\boldsymbol{\delta}\| &\le \frac{m}{L_{total}}
\end{aligned}
$$

The overall robustness bound $\epsilon$ is therefore:

$$
\begin{aligned}
\epsilon &= \frac{m}{\sum_i (r_{R_i} M_{R_i} + a_{R_i}(\mathbf{x}) L_{R_i})} \\
&= \min_{k \ne y} \frac{F_R^{(y)}(\mathbf{x}) - F_R^{(k)}(\mathbf{x})}{\sum_i (r_{R_i} M_{R_i} + a_{R_i}(\mathbf{x}) L_{R_i})}
\end{aligned}
$$

This concludes the proof. ∎

**Proof of Theorem 5.5:** To derive the certified bound for the dual-model, we utilize Theorem 3.5 from (Bai et al., 2024a), which relates the robustness bound of a dual-model to the properties of the robust model. Since $\alpha$ is assumed to be in $[\frac{1}{2}, 1]$, let $\delta \in \mathbb{R}^d$ satisfy $\|\delta\|_p \le r_{\text{Lip},p}^\alpha(x)$. Furthermore, for $i \in [c] \backslash \{y\}$, we have

$$
\begin{aligned}
\sigma \circ & h_y(x + \delta) - \sigma \circ h_i(x + \delta) \\
&= \sigma \circ h_y(x) - \sigma \circ h_i(x) + \sigma \circ h_y(x + \delta) \\
&\quad - \sigma \circ h_y(x) + \sigma \circ h_i(x) - \sigma \circ h_i(x + \delta) \\
&\ge \sigma \circ h_y(x) - \sigma \circ h_i(x) - \text{Lip}_p (\sigma \circ h_y) \|\delta\|_p \\
&\quad - \text{Lip}_p (\sigma \circ h_i) \|\delta\|_p \\
&\ge \sigma \circ h_y(x) - \sigma \circ h_i(x) \\
&\quad - \left( \text{Lip}_p (\sigma \circ h_y) + \text{Lip}_p (\sigma \circ h_i) \right) r_{\text{Lip},p}^\alpha(x) \\
&\ge \frac{1 - \alpha}{\alpha},
\end{aligned}
\tag{11}
$$

where $\alpha$ is the parameter controlling the contribution of the robust model to the dual-model, $\sigma \circ h_y(x)$ represents the output of the robust model for the true class $y$ and $\text{Lip}_p (\sigma \circ h_i)$

denotes the Lipschitz constant of the robust model for class $i$.

Then, the robustness bound of the dual-model can be expressed as:

$$\|\delta\|_p \leq \min_{i \neq y} \frac{\alpha \left(\sigma \circ h_y(x) - \sigma \circ h_i(x)\right) + \alpha - 1}{\alpha \left(\text{Lip}_p \left(\sigma \circ h_y\right) + \text{Lip}_p \left(\sigma \circ h_i\right)\right)}.$$

From the previous proof, we already derived the Lipschitz constant for the robust MoE. Using this result, we can now represent the robustness bound $\epsilon$ for the dual-model as:

$$\epsilon = \min_{k \neq y} \frac{\alpha(F_R^{(y)}(\mathbf{x}) - F_R^{(k)}(\mathbf{x})) + \alpha - 1}{\alpha \sum_i \left(2r_{R_i} + a_{R_i}(\mathbf{x})(L_{R_i}^{(y)} + L_{R_i}^{(k)})\right)}$$

This concludes the proof. ∎

### A.2. Model Architecture Details

Following several recent works (Videau et al., 2024; Chen et al., 2025; He et al., 2024), the default setting is a single MoE layer as a replacement for the classification layer. We further conduct experiments using the architecture proposed by Riquelme et al. (2021), which inserts MoE layers in place of the dense MLPs within transformer blocks. After standard training, the MoE model achieves 90.35% SA, 38.02% RA, 32.16% RA-E, and 64.97% RA-R. These results further support our observation that expert networks are generally more vulnerable to adversarial perturbations than the router. This vulnerability likely stems from the fact that expert networks are deeper and more complex than the router, making them inherently more susceptible to such attacks. Regardless of the specific MoE architecture used, this trend remains consistent: the expert networks exhibit greater vulnerability due to their complexity.

### A.3. Additional Experiments on Robust MoE

**Performance of RT-ER on TinyImageNet Dataset.** For the TinyImageNet dataset (Deng et al., 2009), we adversarially trained an MoE model using ViT-small as the experts. The experimental setup follows the same settings outlined in Section 6. The RA and SA performance curves are shown in Figure 4.

RT-ER achieves noticeably higher RA and demonstrates more consistent performance compared to the AT MoE model. While the AT model exhibits some fluctuations in performance, these are less pronounced than those observed on the CIFAR-10 dataset. This improved stability can be attributed to the use of a pre-trained ViT-small model, which inherently possesses adversarial robustness due to its training on a large and diverse dataset. In summary, RT-ER improves RA by over 5.8% while incurring only a modest 2.2% decrease in SA, highlighting its effectiveness in enhancing the robustness of MoE models.

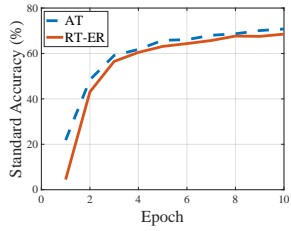 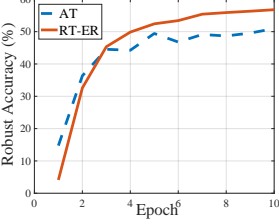

| (a) TinyImageNet SA | (b) TinyImageNet RA |
|---|---|

*Figure 4.* Performance evaluation of AT MoE and RT-ER MoE models with ViT-small on the TinyImageNet test dataset. We report standard accuracy (SA) and robust accuracy (RA) under a 50-step PGD attack, using models trained with a 10-step PGD attack. Our results indicate that RT-ER achieves higher RA and demonstrates greater stability compared to AT MoE.

**Smooth Attack.** Throughout this paper, we adopt the top-1 strategy for expert selection, which is inherently non-differentiable and may cause standard gradient-based attacks to fail. To enable effective robustness evaluation, we employ adaptive attacks by replacing the top-1 selection with a dense smooth approximation. Specifically, we compute a weighted average of expert outputs using the routing weights and generate perturbations $\delta$ based on this smoothed output. This approach ensures that expert selection remains differentiable, allowing gradients to propagate. The results, presented in Table 6, demonstrate that our method, RT-ER, continues to outperform ST and AT in RA.

*Table 6.* Robustness evaluation of standard training (ST), adversarial training (AT), and our method (RT-ER). We adopt a dense smooth approximation to ensure differentiable expert selection. The experiment is conducted on CIFAR-10 using an MoE model with ResNet18 experts.

| CIFAR-10 (Smooth Attack) | | | |
|---|---|---|---|
| Method | SA(%) | RA(%) | RA-E(%) | RA-R(%) |
| ST | 92.14 | 46.02 | **3.11** | 54.67 |
| AT | 79.08 | **51.03** | 73.85 | 78.91 |
| RT-ER | 77.81 | **68.96** | 75.71 | 72.28 |

**Comparison with Trades and AdvMoE.** We compare our method, RT-ER, with TRADES (Zhang et al., 2019) and AdvMoE (Zhang et al., 2023) using an MoE architecture based on ResNet18 on the CIFAR-10 dataset. All experiments are conducted under the same settings for a fair comparison. The results for standard accuracy (SA) and robust accuracy (RA) are presented in Figure 5.

All methods are trained from scratch to ensure consistency. While TRADES and AdvMoE exhibit performance fluctuations similar to those observed in standard adversarial training, RT-ER demonstrates significantly improved stabil-

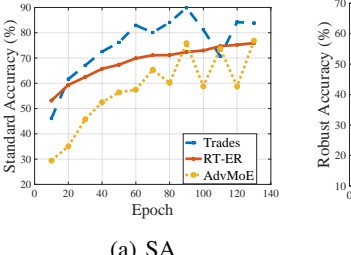
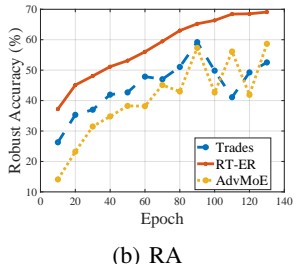

(a) SA         (b) RA

*Figure 5.* Performance comparison of TRADES, AdvMoE, and RT-ER with ResNet18 on the CIFAR-10 test dataset. SA and RA are evaluated under a 50-step PGD attack, using models trained with a 10-step PGD attack. RT-ER achieves higher RA and exhibits greater stability compared to TRADES and AdvMoE. Numerical analysis comparisons are presented in Table 7.

ity. Although RT-ER achieves a marginally higher SA—by less than 1%—compared to TRADES, it achieves a notable 10.42% improvement in RA, underscoring its effectiveness in enhancing robustness. In conclusion, RT-ER is the superior choice for robustifying the MoE layer, delivering both improved robustness and more reliable performance.

*Table 7.* Performance comparison of TRADES, AdvMoE, and RT-ER with ResNet18 on the CIFAR-10 test dataset under PGD and AutoAttack. As AdvMoE is specifically designed for CNN-based MoE, this experiment is conducted using MoE with ResNet18 experts on the CIFAR-10 dataset.

| CIFAR-10 (PGD) | | | | |
| --- | --- | --- | --- | --- |
| Method | SA(%) | RA(%) | RA-E(%) | RA-R(%) |
| Trades | 83.81 | **52.54** | 72.72 | 76.63 |
| AdvMoE | 76.83 | **58.67** | 73.79 | 74.03 |
| RT-ER | 77.81 | **69.09** | 75.71 | 72.28 |
| CIFAR-10 (AA) | | | | |
| Method | SA(%) | RA(%) | RA-E(%) | RA-R(%) |
| Trades | 83.81 | **38.90** | 54.72 | 56.84 |
| AdvMoE | 76.83 | **44.17** | 54.62 | 55.06 |
| RT-ER | 75.92 | **54.36** | 57.86 | 54.84 |

**The Impact of Number of Experts $E$.** We investigate the impact of the number of experts on RT-ER in Table 8. Specifically, we consider three cases: $E \in \{3, 4, 5\}$, using the default CIFAR-10 settings, with the only change being the number of experts. As the number of experts (E) increases, the capacity of the MoE architecture grows. After 130 epochs of training, we observe an approximate 2% improvement in SA, while RA shows only a slight improvement. This is attributed to the sparse MoE and top-1 routing strategy, where only one expert (ResNet18) is activated for each input image. Since each expert is robustified during training and capable of handling classification tasks independently, the SA improvement is more noticeable. These results also demonstrate that RT-ER is adaptable to various

configurations of the MoE architecture.

*Table 8.* Robustness evaluation of RT-ER with varying numbers of experts $E$ on the CIFAR-10 dataset. Increasing E enhances model capacity, leading to improvements in both standard accuracy (SA) and robust accuracy (RA).

| CIFAR-10 | | | | |
| --- | --- | --- | --- | --- |
| E | SA(%) | RA(%) | RA-E(%) | RA-R(%) |
| 3 | 75.85 | **69.03** | 74.79 | 72.42 |
| 4 | 77.81 | **69.09** | 75.71 | 72.28 |
| 5 | 79.77 | **70.00** | 76.51 | 71.86 |

**The Impact of Different $\beta$.** Recall that the loss function $\mathcal{L}_{rob}$ used in the RT-ER method is defined as:

$$\max_{\|\boldsymbol{\delta}\|_p \le \epsilon} \ell_{CE}(F_R(\mathbf{x}+\boldsymbol{\delta}), y) + \beta \cdot \sum_{i=1}^{E} \ell_{KL}(f_{R_i}(\mathbf{x}+\boldsymbol{\delta}), f_{R_i}(\mathbf{x})).$$

We examine the impact of varying $\beta$ on the model's performance, setting $\beta$ to 1, 3, 6, and 9. The results are summarized in Table 9.

*Table 9.* Performance evaluation of RT-ER with different values of $\beta$ in $\mathcal{L}_{rob}$. Metrics reported are standard accuracy (SA), robust accuracy on the entire model (RA), robust accuracy on experts (RA-E), and robust accuracy on the router (RA-R) under a 50-step PGD attack. Results show the influence of $\beta$ on robustness and standard accuracy.

| CIFAR-10 | | | | |
| --- | --- | --- | --- | --- |
| $\beta$ | SA(%) | RA(%) | RA-E(%) | RA-R(%) |
| 1 | 81.68 | 64.13 | 68.67 | 69.64 |
| 3 | 79.91 | 65.92 | 70.75 | 72.05 |
| 6 | 77.81 | 69.09 | 75.71 | 72.28 |
| 9 | 10 | 10 | 10 | 10 |

As shown in the table, increasing $\beta$ leads to a consistent decrease in SA and corresponding increases in RA, RA-E, and RA-R, reflecting improved robustness. However, for $\beta = 9$, all metrics collapse to 10%, indicating random guessing by the model. This phenomenon arises because the second term in $\mathcal{L}_{rob}$, $\sum_{i=1}^{E} \ell_{KL}(f_{R_i}(\mathbf{x} + \boldsymbol{\delta}), f_{R_i}(\mathbf{x}))$, dominates the loss at this value of $\beta$. This term forces experts to approximate their adversarial outputs $f_{R_i}(\mathbf{x}+\boldsymbol{\delta})$ to their initial outputs $f_{R_i}(\mathbf{x})$. Since the MoE is trained from scratch, its initial outputs are meaningless at the beginning of training. Consequently, the model fails to learn effectively, resulting in degenerate performance.

### A.4. Experiments with Pre-trained Dual-Model

In this section, we evaluate the dual-model's performance using pre-trained MoE models under AutoAttack. The results are shown in Figure 6. For CIFAR-10, the attack strength is

set to $\epsilon = 8/255$, while for TinyImageNet, it is $\epsilon = 2/255$. We use a 50-step AutoAttack with a step size of $\epsilon = 2/255$.

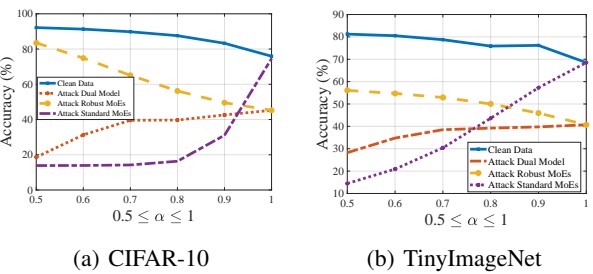

(a) CIFAR-10        (b) TinyImageNet

*Figure 6.* Performance evaluation of the Dual-Model using pretrained MoE models. The Dual-Model is evaluated under AutoAttack, combining a standard MoE (ST) and a robust MoE (RT-ER) from Table 3. The parameter $\alpha$ is varied from 0.5 to 1.0 in steps of 0.1; at $\alpha = 1$, the Dual-Model relies entirely on the robust MoE.

The results indicate that the dual-model's SA decreases while RA increases as $\alpha$ grows. This trend aligns with the structure of the dual-model: as the robust MoE's contribution increases (i.e., higher $\alpha$), RA improves because the robust MoE dominates the final predictions, mitigating the effect of adversarial perturbations on the standard MoE. These findings are consistent with the model performance under PGD attacks, as shown in Section 6.3.

### A.5. JTDMoE Experimental Results

**Effectiveness of JTDMoE Under Different $\alpha$.** We demonstrated the effectiveness of JTDMoE when $\alpha = 0.7$ in Section 6. Here, we evaluate the performance of the dual-model for different values of $\alpha = 0.5, 0.6, 0.7, 0.8$ to show that JTDMoE remains effective across a range of $\alpha$. The standard MoE and robust MoE utilize ResNet18 as the experts. All other settings remain consistent with those in Appendix A.3. The results are illustrated in Figure 7.

From Figure 7, we observe that SA consistently improves under different $\alpha$ during the training process, while RA remains stable or improves slightly. When $\alpha$ is small—indicating that the robust MoE contributes less to the dual-model—the RA continues to increase, showcasing the model's potential for robustness. These results highlight JTDMoE's ability to improve the dual-model's performance, particularly in terms of SA, while maintaining robust accuracy. This demonstrates the effectiveness and adaptability of JTDMoE across varying values of $\alpha$.

### A.6. ViT Experimental results on ImageNet

**Performance of RT-ER on the ImageNet Dataset.** To evaluate the scalability of RT-ER to large-scale models and datasets, we conduct experiments using ViT on the Ima-

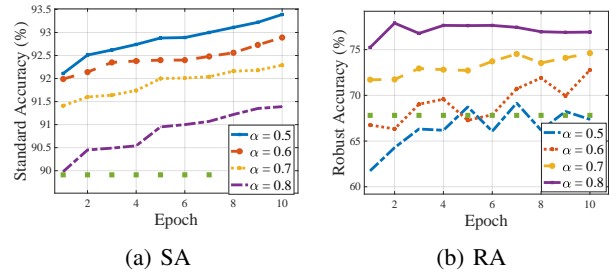

(a) SA                (b) RA

*Figure 7.* Performance evaluation of JTDMoE under different values of $\alpha$ on the CIFAR-10 test dataset. We use the dual-model performance at $\alpha = 0.7$ as the baseline, depicted by green squares in this figure. Standard accuracy (SA) and robust accuracy (RA) under a 50-step PGD attack are reported, with models trained using a 10-step PGD attack. The results demonstrate that JTDMoE maintains effectiveness across different $\alpha$ values.

geNet dataset (Deng et al., 2009). The experimental setup follows the protocol described in Section 6, and the results are summarized in Table 10. RT-ER consistently outperforms both AT and TRADES across all evaluation metrics—SA (+8%), RA (+12%), RA-E (+1%), and RA-R (+0.6%)—demonstrating its effectiveness and efficiency in improving the robustness of MoE models at scale.

*Table 10.* Robustness evaluation of AT, TRADES, and RT-ER. RT-ER achieves a substantial improvement of approximately 12% in RA and 8% in SA compared to conventional adversarial training. The evaluation is conducted on the ImageNet dataset using a MoE model with ViT-based experts.

| ImageNet | | | | |
|---|---|---|---|---|
| Method | SA(%) | RA(%) | RA-E(%) | RA-R(%) |
| AT | 60.32 | 44.64 | 43.06 | 70.24 |
| TRADES | 61.94 | 45.54 | 43.75 | 70.37 |
| RT-ER | 68.38 | 56.16 | 44.99 | 70.82 |

