# OpenReview forum: "Optimizing Robustness and Accuracy in Mixture of Experts: A Dual-Model Approach"
_ICML.cc/2025/Conference — ICML 2025 poster_

### Official Review · Reviewer_nXYP · 2025-02-21

**Overall Recommendation:** 3

**Summary:**

This paper proposes a novel adversarial training algorithm to improve the robustness of MoE models based on pilot studies on MoE attacks. They further interpolate the robust MoE and the unrobust MoE by linear interpolation to balance clean accuracy and robust accuracy.

**Claims And Evidence:**

I am concerned about the claims made in Section 4 and 5. See questions for details.

**Essential References Not Discussed:**

The reference is sufficiently discussed.

**Experimental Designs Or Analyses:**

I have major concerns regarding the design of experiments. See questions.

**Methods And Evaluation Criteria:**

The method mostly make sense. See concerns regarding its implementation, evaluation and intuitions in questions.

**Other Comments Or Suggestions:**

Eq 2 has typos.

\icmltitlerunning is not properly set.

**Other Strengths And Weaknesses:**

The main strength of this work is to consider a novel perspective of MoE models. However, since MoE models are more commonly used in LLMs but not classical problems, maybe it's better to evaluate the proposed methods in LLMs as well.

**Questions For Authors:**

1. In Section 4.1, Why RA-E attacks are even better than attacking the full MoE, i.e., RA? This usually suggests that the applied attack is too weak (or badly applied) for the full MoE model. This might also explain the bad performance of adversarial training in Section 4.2. Further, Fig 2 shows the standard accuracy for AT is higher and the robust accuracy is lower, indicating a bad attack as well. I took a brief read on the code, and surprisingly the authors use a self-implemented AutoAttack, while the official library is easy-to-install and publicly available at https://github.com/fra31/auto-attack. This makes me question the effectiveness of the implementation. Further, the PGD attack code simply runs for a fixed number of iterations and takes the end value, while it should take the highest loss point during the full trajectory.

2. In Section 4.1, RA-E and RA-R attacks a changed formulation of MoE; is the accuracy here still evaluated w.r.t. the original MoE model? I can hardly believe that since RA-E attack assumes constant router score during attack, this remains effective when the router scores are changed by the adversarial attack. More details should be provided.

3. Eq 2 looks like applying AT on the full MoE and including TRADE on the second expert. This should be discussed or unified. Currently, there seems to be no evidence why such choice is better than (i) use AT on the second expert and (ii) use TRADE on the full MoE.

4. Section 4.2 changes MoE to top-1 and top-2 routers. Could the authors include the results of the full MoE model, which is used by other experiments?

5. Line 274, why does $\alpha \ge 0.5$ guarantee robustness of the dual-model?

6. According to Line 280, robustifying every expert is essential. Why does RT-ER only improve the second expert but not all experts?

7. The dual model effectively interpolates a clean model and a robust model; can't we simply train the robust model with $\alpha L_{\text{clean}} + (1-\alpha) L_{\text{rob}}$ to get the same effect? This is more common when one wants to do interpolation. In addition, the dual model is twice the size of a single model, thus the comparison in Table 3 is unfair.

**Relation To Broader Scientific Literature:**

This work combines adversarial robustness and MoE models. While both are extensively studied, I am not aware of prior studies on the adversarial robustness of MoE.

**Theoretical Claims:**

The theorems are intuitive but I do not check their formal proof.

---

> ### Author Rebuttal · Authors · 2025-03-31
>
> Thank you for raising the seven insightful questions. We appreciate your thorough review and address each concern in detail below.
>
> 1. ## Why Attack is not Weak?
>
>    ​	We compared our self-implemented AutoAttack with the official library. The perturbations generated by both methods are identical, and the results on the CIFAR-10 match. Second, regarding the concern that our PGD attack takes the end value, we note that many other studies adopt similar attack settings, such as [1,2]. Additionally, we conducted further experiments on TinyImageNet, both by using the final value and by selecting the highest loss point. The attack outcomes were nearly identical, and the loss values remained smooth in the later iterations.
>
> 2. ## Why RA-E Attack Remains Effective?
>
>    ​	We would like to clarify that our evaluation assesses the vulnerability of both the router and experts by measuring accuracy relative to MoE. Regarding your concern about the RA-E attack’s effectiveness under varying router scores, we emphasize that this attack primarily targets the experts. As a result, the perturbations remain largely independent of the router’s behavior. Empirically, our experiments show that in **98%** of cases, attacked images are routed to the same expert(s), ensuring that the RA-E attack remains effective.
>
> 3. ## Why Using KL Divergence in Eq.2?
>
>    ​	The goal of our paper is to explore a potential approach to further enhance the robustness of MoE. Regarding the use of KL divergence in the second term of Eq. 2, we chose this formulation because applying AT to the second expert would force the additional expert’s predictions toward the ground truth, potentially leading to overfitting.
>
>    ​	We conducted experiments comparing our method against two alternative formulations you suggested. Our results showed that RT-ER achieved similar performance to (ii) while improving RA by 2% compared to (i).
>
> 4. ## Inclusion of Full MoE Results
>
>    ​	In Section 4.2, we compare the performance of MoEs using top-1 and top-2 routing strategies to demonstrate that our method generalizes across different routing mechanisms.
>
>    ​	To further address your request, we provide additional results for a Dense MoE and an MoE using a top-3 router on the CIFAR-10:
>
>    | Method | SA(%) | RA(%) | RA-E(%) | RA-R(%) |
>    | ------ | ----- | ----- | ------- | ------- |
>    | Dense  | 79.23 | 70.50 | 76.51   | 72.62   |
>    | Top-3  | 79.18 | 70.11 | 76.24   | 72.48   |
>
> 5. ## Why $\alpha$≥0.5 Guarantees Robustness of the Dual-Model?
>
>    ​	We provide theoretical evidence supporting the conclusion that$\alpha \geq 0.5$ is necessary to guarantee the robustness of the dual-model.
>
>    ​	Eq. (9) establishes a bound on the certified robustness radius:
>    $$
>    \lVert \delta \rVert_p \leq \epsilon = \min\limits_{k \neq y} \frac{\alpha \left(F_R^{(y)}(x) - F_R^{(k)}(x)\right) + \alpha - 1}{\alpha \sum_i \left(2 r_{R_i} + a_{R_i}({x}) \left(L_{R_i}^{(y)} + L_{R_i}^{(k)}\right) \right)}.
>    $$
>    ​	In the numerator of Eq. (9), $\alpha$ ranges from 0 to 1, and the maximum value of $\left(F_R^{(y)}(\mathbf{x}) - F_R^{(k)}(\mathbf{x})\right)$ is 1. If $\alpha$ is smaller than 0.5, the numerator becomes negative,making the certified robustness radius undefined. Therefore, we conclude that $\alpha \geq 0.5$ is a necessary condition to ensure the robustness of the dual-model.
>
> 6. ## Why only Robustifying the Second Expert?
>
>    ​	Directly robustifying all experts simultaneously would be computationally expensive and inefficient, making MoEs less appealing for large-scale applications. To balance efficiency and robustness, we propose RT-ER. For each input, RT-ER additionally robustifies an **expert not selected** by the router. Since the router dynamically selects different experts during training, the additional expert chosen for robustification also varies over time. This iterative process enables **RT-ER to progressively improve the robustness of the entire expert network** without significantly increasing computational costs.
>
> 7. ## Why Not Simply Train a Robust Model?
>
>    ​	Compared to simply training a robust model with $(1 - \alpha) L_{clean} + \alpha L_{rob}$, our method offers two key advantages:
>
>    1. **Better SA-RA Tradeoff**: The dual-model strategy in JTDMoE provides a better balance between SA and RA. Simply training often leads to conflicting gradients, limiting performance. We compare JTDMoE with a MoE of 8 experts trained using $\alpha = 0.7$ and the combined loss. On CIFAR-10, JTDMoE achieved 92.29% SA and 74.62% RA, whereas the simply trained model reached only 87.48% SA and 67.45% RA. This shows that JTDMoE achieves a significantly better tradeoff.
>    2. **Reduced Training Costs**: Our approach eliminates the need for adversarial training in the standard MoE.
>
>    [1] Zhang Robust mixture-of-expert training for convolutional neural networks.
>
>    [2] Bai Improving the accuracy-robustness trade-off of classifiers via adaptive smoothing.

---

> > ### Comment · Reviewer_nXYP · 2025-04-02
> >
> > Dear authors,
> >
> > Thanks for the rebuttal. It clears most of my concerns except one:
> >
> > Q1: I appreciate the comparison with the official code. However, this does not explain why RA-E is better than attacking the full MoE. Could you discuss this?
> >
> > Depending on the answers to my other questions and the new ImageNet results in the reply to Reviewer vMs4, I decide to raise my score to 3.

---

> > > ### Author Response · Authors · 2025-04-02
> > >
> > > Thank you for your follow-up question and for raising your score! We’re happy to address your question and provide further clarification.
> > >
> > > When an adversary focuses solely on attacking the experts (RA-E), the perturbation can directly target the most vulnerable part of the system without interference from other components. When the attack targets only the experts, the perturbation is computed independently of the router. Its sole objective is to degrade the performance of the expert(s) that the router selects, with no other experts available to counteract the attack.
> > >
> > > When the full MoE is attacked (RA), the adversarial gradient is influenced by both the experts and the router. The router may adaptively shift to activate a different expert in response to the perturbation (This also needs more attack resources as the router tends to be more robust with its simpler structure). As the perturbation evolves, the target expert(s) may change due to the router's dynamic selection. This variability means that the adversary must continually adapt its perturbation to affect different experts, reducing the overall attack efficiency.
> > >
> > > In summary, attacking only the experts yields a lower robust accuracy (RA-E) because it isolates and exploits the inherent vulnerability of the selected experts without the mitigating effect of the robust router.

---

### Official Review · Reviewer_vMs4 · 2025-03-10

**Overall Recommendation:** 1

**Summary:**

The paper proposes:
1. A loss function that specifically enhance the robustness of experts in MoE architecture;
2. A dual-model strategy for robustness-accuracy trade-off;
3. A joint-training strategy for dual-model.
to enhance the adversarial robustness of MoE model.
Experiments are conducted on CIFAR10 and TinyImageNet.

**Claims And Evidence:**

The analysis of vulnerability of different part of MoE model is clear and convincing.

**Essential References Not Discussed:**

None

**Experimental Designs Or Analyses:**

The experiment is conducted on small-scaled datasets, such as CIFAR10 and TinyImageNet. But the MoE architecture is designed for large-scaled datasets, where needs large model but control the computational overhead. Therefore, I think the author should contains more experiment results on larger-scaled datasets, like ImageNet-1K or ImageNet-21K.

**Methods And Evaluation Criteria:**

The proposed method contains three parts, but in some degree, more like a combination of different methods. And these robustness enhance methods, like aligning outputs with KL-loss and mixing clean and adversarial outputs, are similar with existing methods for classical neural network. The above factors decline the novelty of the proposed method.

**Other Comments Or Suggestions:**

None

**Other Strengths And Weaknesses:**

None

**Questions For Authors:**

None

**Relation To Broader Scientific Literature:**

None

**Theoretical Claims:**

None

---

> ### Author Rebuttal · Authors · 2025-03-31
>
> Thank you for your comments.
>
> ## Contributions of Our Method
>
> We appreciate the reviewer's comment. We would like to point out that aligning outputs with KL-loss and mixing clean and adversarial outputs are classical loss design techniques used in most of the adversarial training papers. We would like to clarify that our contributions are novel and **specifically tailored to the unique challenges of MoE architectures**. In particular:
>
> **MoE-Specific Vulnerability Analysis:** Our work is grounded in the empirical and theoretical observation that expert networks in MoEs are significantly more susceptible to adversarial attacks than the router. This insight is critical because, unlike standard neural networks, the MoE framework relies on dynamic routing where the experts' outputs directly affect the final prediction. Our contribution lies in isolating and addressing this vulnerability, which is not encountered in traditional architectures.
>
> **RT-ER – A Targeted Robustification Strategy:** While methods such as aligning outputs with KL divergence have been explored in classical settings, our RT-ER method is designed specifically for MoEs. For each input, we robustify an expert that is not selected by the router, and because the router's selection changes during training, this process iteratively reinforces the robustness of the entire expert network. This dynamic and efficient strategy is uniquely adapted to the MoE structure and its operational dynamics, setting it apart from standard adversarial training methods.
>
> **Dual-Model Strategy (JTDMoE):** Our dual-model strategy is not a mere combination of existing techniques but a novel design that integrates a standard MoE and a robust MoE in a unified framework. This approach allows us to achieve a favorable balance between clean and robust accuracy while maintaining efficiency. Unlike typical ensemble methods in classical networks, our dual-model is trained with a specifically designed bi-level training process and is accompanied by a rigorous theoretical robustness bound (Theorem 5.5) that quantifies how the interplay between the two models contributes to overall robustness.
>
> **Theoretical Contributions:** We provide new certified robustness bounds for both the full MoE model and the dual-model setup, offering insights into how individual components (especially the experts) impact overall robustness. These theoretical guarantees are tailored to the dynamics of MoEs and serve as a foundation for our proposed training strategies, further reinforcing the novelty of our contributions.
>
> In summary, our method addresses core challenges in MoE training (namely robustness, performance, and efficiency) by proposing tailored strategies (RT-ER and JTDMoE) and accompanying theoretical guarantees that are specifically designed for the MoE setting. These contributions collectively advance the state-of-the-art in adversarial robustness for MoEs and are not simply an assembly of existing techniques from classical neural networks.
>
> ## Results on Large-Scaled Datasets
>
> Thank you for your comments. Our work primarily focuses on investigating the robustness of MoE. To evaluate our proposed method, we adopt the **ViT + TinyImageNet** and **ResNet + CIFAR-10** settings—both of which are widely used benchmarks in robust MoE research. Notably, this evaluation setup has also been employed in recent studies, such as Lin [1] and Zhang [2], further supporting its relevance and acceptance within the community. Although our current experiments focus on CIFAR-10 and TinyImageNet, our method is designed to be scalable. The computational cost of robustifying one additional expert is minimal relative to the overall training budget, ensuring that the approach can be applied efficiently to larger datasets and more complex models.
>
> To further strengthen our empirical validation, we now additionally provide results for our proposed method using a **ViT model on the full ImageNet dataset**. The results are summarized as follows:
>
> | Method               | **SA(%)**      | **RA(%)**      | **RA-E(%)**    | **RA-R(%)**    |
> | -------------------- | -------------- | -------------- | -------------- | -------------- |
> | RT-ER                | 68.38$\pm$0.17 | 56.16$\pm$0.14 | 44.99$\pm$0.16 | 70.82$\pm$0.13 |
> | Adversarial Training | 60.32$\pm$0.15 | 44.64$\pm$0.14 | 43.06$\pm$0.17 | 70.24$\pm$0.14 |
> | Trades               | 61.94$\pm$0.12 | 45.54$\pm$0.17 | 43.75$\pm$0.13 | 70.37$\pm$0.11 |
>
> These results demonstrate that our method (RT-ER) consistently outperforms AT and Trades on the ImageNet. Notably, RT-ER achieves **12% (10%) improvement in RA** and **8% (6%) improvement in SA** compared with AT (Trades). This underscores the scalability and effectiveness of our approach, even on large-scale vision benchmarks like ImageNet.
>
> [1] Lin F et al. Towards Robust Vision Transformer via Masked Adaptive Ensemble.
>
> [2] Zhang Y et al. Robust mixture-of-expert training for convolutional neural networks.

---

### Official Review · Reviewer_K985 · 2025-03-17

**Overall Recommendation:** 3

**Summary:**

The paper studies the adversarial robustness of mixture-of-experts (MoE) models in detail, investigating the susceptibility to adversarial attacks to both the router and the experts modules. Under some assumptions, the paper proofs that the perturbation on the entire model can be decomposed as the sum of the perturbations on the router inputs, and the experts inputs. Based on these, one can bound the Lipschitz constant of the entire MoE model.

The authors suggest the use of a dual model to improve the adversarial robustness, which is comprised by a model trained purely on a classification task, and a model trained with adversarial training. The paper shows that once can derive a certified robustness bound from the dual-model as well.

The paper presents experimental results on CIFAR-10 and TinyImageNet, showing that the proposed method performs better than vanilla adversarial training, using a ResNet-based MoE model.

### Update after the rebuttal
I thank the authors for clarifying in which scenarios Assumption 5.3 is realistic. I agree with their analysis in that regard.

Regarding my two other concerns, the authors mention that they follow practices used in previously published papers (i.e. the choice of TinyImageNet, and the use of MoE models with as single MoE layer at the end). This argument is a bit weak, in my opinion. The fact that previous works have been published with a suboptimal evaluation method (in my humble opinion), doesn't mean that we should continue doing that. Nevertheless, the authors ran some additional experiments on the full ImageNet and using full MoE models (i.e. replacing the MLP in ViT with a MoE model), strengthening the evidence supporting their proposed method.

Given this, I have increased my score and I'm (slightly) leaning towards acceptance.

**Claims And Evidence:**

The paper successfully identifies how each component of an MoE model affects the robustness, theoretically (deriving an upper bound for each term) and empirically (observing that the models used in the experiments are more susceptible to attacks in the expert modules).

The proposed dual-model approach, and the joint training strategy to train them, obtain very successful results when compared to standard adversarial robustness training, both in terms of accuracy on the clean data and under adversarial attacks.

**Essential References Not Discussed:**

Given my concerns about the (in)validity of the theoretical assumptions with non-continuous routers, I would suggest the authors refering to papers that try to amend this, such as "From Sparse to Soft Mixtures of Experts" by Puigcerver et al (2023), and the more recent "ReMoE: Fully Differentiable Mixture-of-Experts with ReLU Routing" by Zhang et al. (2024).

It's probably too much to ask, but it would be interesting to repeat some of the analysis with one of these MoE approaches, to check if the findings still hold.

**Experimental Designs Or Analyses:**

The experimental design is appropriate for the paper, in terms of evaluation protocol.

However, the choice of the models used (a small ResNet with a single MoE linear layer in the classification head), and the datasets in which the experiments were conducted (CIFAR10 and TinyImageNet) raises some concerns about the relevance of the experiments.

Compare this with (for instance) Puigcerver et al. (2022), which the paper refers to as one of the first works studying the adversarial robustness of MoE models, that used MoEs based on Vision Transformers, and trained on full ImageNet.

In addition to size of the models and datasets itself, there's a key distinction: this work uses only a single MoE layer _as a replacement of the classification layer_, while modern state-of-the-art transformer-based MoE models place MoEs as a replacement of the dense MLPs inside the transformer blocks, and not at the classification layer. This might shift completely one of the key observations from the paper: the fact that the expert modules are more susceptible to adversarial attacks, might be due to the fact that they directly affect the model's output.

**Methods And Evaluation Criteria:**

The evaluation criteria follows the standard one used in adversarial robustness works. Namely, studying the accuracy both on clean data and adversarially constructed inputs with standard methods such as PGD and AutoAttack.

However, see the comments in "Experimental Designs Or Analyses" regarding my concerns with experimentation and the conclusions that one can draw from these.

**Other Comments Or Suggestions:**

The tables feel a little bit too clutered with some many horizontal / vertical bars. I would suggest tidying them up a bit.

**Other Strengths And Weaknesses:**

Strengths:
- The paper is well motivated, structured, and written.
- The theortetical proofs in the appendix easy to follow.
- The proposed methods seem quite easy to implement, which can potentially widen the adoption of the proposed approach.
- The paper contains ablation experiments tuning the hyperparameters of the dual-model strategy.

**Questions For Authors:**

None.

**Relation To Broader Scientific Literature:**

The key contributions are very relevant to the MoE community in computer vision and adversarial robustness. The paper does a good job referencing the main relevant papers from each topic, and making the relationship among them clear.

**Theoretical Claims:**

I checked the proofs for both theorems 5.4 and 5.5. I could not spot any problem in the theorem themselves, but I do have a concern regarding one of the assumptions. In particular, the part relative to the router in Assumption 5.3 doesn't hold in real scenarios (including the ones in the paper's experiment), I believe.

The assumption states that the $i$-th output of the router is Lipschitz continuous with  $\|a_{R_i}(\mathbf{x} + \mathbf{\delta}) - a_{R_i}(\mathbf{x})\| \leq r_{R_i} \|\|\mathbf{\delta}\|\|_p$. However, this assumption is not true for typical sparse MoEs, where $a(\mathbf{x})_i = \text{top}_k (\text{softmax}(W \mathbf{x}))$, at least not for all $\mathbf{x}$.

Consider a top-1 MoE, and a point $\mathbf{x}$ that lies at a distance $\leq \epsilon$ of the decision boundary between the top-2 experts. Without loss of generality, let's assume that these are experts with indices 1 and 2, respectively, and let's assume that the top-1 routing weight is $a(\mathbf{x})_1 = \rho$. Due to discontinuity of standard sparse MoEs, no matter how small I pick $\epsilon$, there exists a perturbation $\mathbf{\delta}$ such that:
- $a(\mathbf{x} + \mathbf{\delta})_1\approx \rho$, and $a(\mathbf{x} + \mathbf{\delta})_2 = 0$
- $a(\mathbf{x} - \mathbf{\delta})_1 = 0$, and $a(\mathbf{x} - \mathbf{\delta})_2  \approx \rho$

Note that $\rho$ can change depending on the exact definition of the router, but in any case it will be $\rho > \frac{1}{n}$, where $n$ is the total number of experts. Thus, completely unrelated to (and much bigger than) $\epsilon$.

Essentially, $\mathbf{\delta}$ is a perturbation that moves the input perpendicular to the boundary between expert 1 and expert 2, closer to one or the other depending on the direction.

---

> ### Author Rebuttal · Authors · 2025-03-31
>
> We appreciate your thoughtful feedback and the opportunity to clarify and strengthen our submission. Below, we respond to each of your concerns in detail.
>
> ## A concern regarding assumption 5.3 not holding in real practice
>
> We would like to clarify that Assumption 5.3 is reasonable and holds in several practical scenarios. Specifically, we outline three cases where this assumption is satisfied:
>
> ​	1.**Sparse MoE after robust training**: Robust training techniques typically encourage large expert score margins. In our setting, a large margin in the expert scores implies that the differences between the scores of the top-k experts are well separated from the rest. This inherent separation makes it difficult for adversarial perturbations to change the top-k set within a realistic $\delta$-ball. In this situation, the router becomes locally stable and consistently selects the same expert(s) for both the clean input $x$ and the perturbed input $x + \delta$, thereby satisfying Assumption 5.3.
>
> ​	2.**Dense MoE**: In a dense MoE [6], all experts are activated for each input, and the routing function is continuous. Thus, Assumption 5.3 naturally holds.
>
> ​	3.**Soft MoE**: Similarly, soft MoE [5] use a continuous routing function with soft assignments. This inherent continuity ensures that small changes in the input lead to small changes in the routing outputs, thereby maintaining stability and satisfying Assumption 5.3.
>
> We have added additional discussion and clarification in the revised version of the manuscript to further address this point.
>
> ## Require large model (ViT) and dataset (ImageNet)
>
> We would like to clarify that we have already included results using a ViT model on the TinyImageNet (see **Table 3** and **Figure 3**). Our work focuses on investigating the robustness of MoEs, and we adopt the ViT + TinyImageNet setting to evaluate our method. This setting has also been adopted in prior studies on the adversarial robustness of MoE, such as [7], making it a widely accepted benchmark for robust MoE research.
>
> To further strengthen our empirical evaluation, we now additionally provide results for our proposed RT-ER using a ViT model on ImageNet. In particular, we observe a substantial improvement of approximately **12% in RA and 8% SA** compared with conventional adversarial training. The results are summarized below:
>
> | **Method** | **SA(%)**      | **RA(%)**      | **RA-E(%)**    | **RA-R(%)**    |
> | ---------- | -------------- | -------------- | -------------- | -------------- |
> | RT-ER      | 68.38$\pm$0.17 | 56.16$\pm$0.14 | 44.99$\pm$0.16 | 70.82$\pm$0.13 |
> | AT         | 60.32$\pm$0.15 | 44.64$\pm$0.14 | 43.06$\pm$0.17 | 70.24$\pm$0.14 |
> | Trades     | 61.94$\pm$0.12 | 45.54$\pm$0.17 | 43.75$\pm$0.13 | 70.37$\pm$0.11 |
>
> ## A Key Distinction in MoE Architecture in Experimental Designs
>
> Thank you for your comments. We would like to clarify that our primary focus is to investigate the *fundamental vulnerability* of MoE architectures to adversarial attacks in the context of **image classification**. Our architectural setup aligns with several recent works that adopt a similar design [2-4], allowing us to pinpoint the vulnerabilities of the experts without interference from additional layers. Importantly, the RT-ER and the JTDMoE strategies we proposed are designed to be agnostic to the underlying MoE integration scheme. We would also like to point out that the observation “the expert modules are more susceptible to adversarial attacks” is *independent of the specific placement* of the MoE layers—whether integrated within the classification head or embedded inside transformer blocks.
>
> To support this claim, we have verified that our findings remain valid under the architecture proposed by Riquelme et al. [8], which replaces the MLP block with a MoE layer. After standard training, the MoE achieved **90.35% SA, 38.02% RA, 32.16% RA-E, and 64.97% RA-R**. Expert networks, being deeper and more complex than the router, are inherently more vulnerable to adversarial perturbations. Even though the experts do not directly produce the final output, their susceptibility can still impact the overall model, as adversarial effects propagate through subsequent layers. These results confirm that the insights and conclusions derived from our analysis are broadly applicable across diverse MoE integration strategies.
>
> [1] Puigcerver et al. On the adversarial robustness of mixture of experts.
>
> [2] Videau et al. Mixture of Experts in Image Classification: What's the Sweet Spot?
>
> [3]Chen et al. Heterogeneous Mixture of Experts for Remote Sensing Image Super-Resolution.
>
> [4] He et al. Mixture-of-experts for semantic segmentation of remoting sensing image.
>
> [5] Puigcerver et al. From sparse to soft mixtures of experts.
>
> [6] Zhang Dense vision transformer compression with few samples.
>
> [7] Lin et al. Towards Robust Vision Transformer via Masked Adaptive Ensemble.
>
> [8] Riquelme et al. Scaling vision with sparse mixture of experts.

---

### Decision · Program_Chairs · 2025-05-01

**Decision:**

Accept (poster)

**Comment:**

This paper proposes an adversarial training method to improve the robustness of MoE models. The approach uses linear interpolation between robust and non-robust MoE models to balance clean and robust accuracy. Reviewers raised some concerns about the novelty of the paper and the significance of its theoretical contributions. They also noted that more results on larger-scale datasets are needed. In the rebuttal, the authors ran additional experiments, which strengthened the empirical evidence.